# Object-Category Aware Reinforcement Learning

**Qi Yi**[1,2,3]  **Rui Zhang**[2,3]  **Shaohui Peng**[2,3,4]  **Jiaming Guo**[2,3,4]
**Xing Hu**[2]  **Zidong Du**[2,3]  **Xishan Zhang**[2,3]  **Qi Guo**[2]  **Yunji Chen**[2,4 †]

[1]University of Science and Technology of China
[2]SKL of Processors, Institute of Computing Technology, CAS
[3]Cambricon Technologies  [4]University of Chinese Academy of Sciences

`yiqi@mail.ustc.edu.cn, {zhangrui, pengshaohui18z, guojiaming18s,`
`huxing, duzidong, zhangxishan, guoqi, cyj}@ict.ac.cn`

## Abstract

Object-oriented reinforcement learning (OORL) is a promising way to improve the sample efficiency and generalization ability over standard RL. Recent works that try to solve OORL tasks without additional feature engineering mainly focus on learning the object representations and then solving tasks via reasoning based on these object representations. However, none of these works tries to explicitly model the inherent similarity between different object instances of the same category. Objects of the same category should share similar functionalities; therefore, the category is the most critical property of an object. Following this insight, we propose a novel framework named Object-Category Aware Reinforcement Learning (OCARL), which utilizes the category information of objects to facilitate both perception and reasoning. OCARL consists of three parts: (1) Category-Aware Unsupervised Object Discovery (UOD), which discovers the objects as well as their corresponding categories; (2) Object-Category Aware Perception, which encodes the category information and is also robust to the incompleteness of (1) at the same time; (3) Object-Centric Modular Reasoning, which adopts multiple independent and object-category-specific networks when reasoning based on objects. Our experiments show that OCARL can improve both the sample efficiency and generalization in the OORL domain.

## 1 Introduction

Reinforcement Learning (RL) has achieved impressive progress in recent years, such as results in Atari [24] and Go [28] in which RL agents even perform better than human beings. Despite its successes, conventional RL is also known to be of low sample efficiency [8] and poor generalization ability [2]. Object-oriented RL (OORL) [10; 6; 23; 17; 39] is a promising way to deal with these limitations. Inspired by studies in cognitive science [25] that objects are the basic units of recognizing the world, OORL focuses on learning the invariance of objects' functionalities in different scenarios to achieve better generalization ability.

In OORL, the agent's observation is a set of object representations, and the task can be solved via reasoning based on these objects. Most previous works [10; 17; 6] in OORL employ hand-crafted object features given in advance, which require human expertise and result in low generality, limiting the applications of these methods. Recent works [39; 38; 37] try to broaden the scope of OORL's applications by avoiding using additional feature engineering. Their works can be divided into either top-down approaches [39] which purely rely on reward signals to ground representations to objects, or bottom-up approaches [38; 37] which utilize unsupervised object discovery (UOD) technologies to provide structural object representations.

---

† Corresponding author.

However, none of these works [39; 38; 37] tries to *explicitly* model the inherent similarity between different object instances of the same *category* when reasoning based on objects. In most cases, objects of the same category should share the same functionalities, such as obeying similar behaviour patterns or having the same effects when interacting with other objects. Therefore, an intelligent agent should recognize an object's category to be aware of its functionalities. Since objects from different categories differ in their functionalities, the objects' functionalities can also be separately modelled (according to their categories), leading to modularity which is beneficial for generalization [4; 5; 36] in general.

Following the above insight, we propose a framework named **O**bject-**C**ategory **A**ware **R**einforcement **L**earning (OCARL), which utilizes the category information of objects to facilitate both perception and reasoning. OCARL consists of three parts: (1) category-aware UOD, which can automatically discover the objects *as well as their corresponding categories* via UOD and unsupervised clustering. (2) **O**bject-**C**ategory **A**ware **P**erception (OCAP), a perception module that takes the object category information obtained in (1) as an *additional* supervision signal. OCAP can encode the category information into representations and is also robust to the incompleteness of UOD at the same time. (3) **O**bject-**C**entric **M**odular **R**easoning (OCMR), a reasoning module that adopts multiple independent and object-category-specific networks, each of which processes the object features of the same corresponding category. Such a modular mechanism can enhance further the generalization ability of OCARL. Our experiments show that OCARL can improve the sample efficiency on several tasks and also enable the agent to generalize to unseen combinations of objects where other baselines fail.

## 2 Related Works

**Unsupervised Object Discovery** Unsupervised object discovery (UOD) methods try to automatically discover objects without additional supervision. Roughly speaking, there are two main categories in this area: spatial attention models and spatial mixture models. Spatial attention models such as SCALOR [16] and TBA [14] explicitly factorize the scene into a set of object properties such as position, scale and presence. Then these properties are utilized by a spatial transformer network to select small patches in the original image, which constitute a set of object proposals. These methods can deal with a flexible number of objects and thus are prominent in the UOD tasks. Spacial mixture models such as Slot Attention [21], IODINE [9] and MONet [1] decompose scenes via clustering pixels that belong to the same object (often in an iterative manner). However, they often assume a fixed maximum number of objects and thus cannot deal with a large number of objects.

SPACE [20], a UOD method used in this paper, combines both the spatial attention model and spatial mixture model. The spatial attention model extracts objects from the scene, whereas the spacial mixture model is responsible for decomposing the remaining background. Such a combination enables SPACE to distinguish salient objects from relative complex backgrounds.

**Object-Oriented Reinforcement Learning** Object-oriented reinforcement learning (OORL) has been widely studied in RL community. There are many different structural assumptions on the underlying MDP in OORL, such as relational MDP [10], propositional object-oriented MDPs [6], and deictic object-oriented MDP [23]. Although these assumptions differ in detail, they all share a common spirit that the state-space of MDPs can be represented in terms of objects. Following these assumptions, [3] proposes object-focused Q-learning to improve sample efficiency. [17] learns an object-based casual model for regression planning. [31] improves the generalization ability to novel objects by leveraging relation types between objects that are given in advance. Although these works have been demonstrated to be useful in their corresponding domain, they require explicit encodings of object representations, which limits their applications.

There are other lines of work in OORL such as COBRA [33], OODP [40], and OP3 [30] that try to solve OORL tasks in an end-to-end fashion. Most of these works fall into a model-based paradigm in which an object-centric model is trained and then utilized to plan. Although these methods are effective in their corresponding domains, in this paper, we restrict ourselves to the *model-free* setting, which can generally achieve better asymptotic performance than model-based methods [32; 12].

The most related works should be RRL [39] and SMORL [38]. RRL proposes an attention-based neural network to introduce a relational inductive bias into RL agents. However, since RRL relies purely on this bias, it may fail to capture the objects from the observations. On the other hand,

SMORL utilizes an advanced UOD method to extract object representations from raw images, and then these representations are directly used as the whole observations. However, SMORL adopts a relatively simple reasoning module, which limits its applications in multi-object scenarios.

## 3 Method

OCARL consists of three parts: (1) category-aware unsupervised object discovery (category-aware UOD) module, (2) object-category aware perception (OCAP), and (3) object-centric modular reasoning (OCMR) module. The overall architecture is shown in Figure 1. We will introduce (1)(2)(3) in Section 3.1, Section 3.2, and Section 3.3 respectively.

### 3.1 Category-Aware Unsupervised Object Discovery

**Unsupervised Object Discovery**    In this work, SPACE [20] is utilized to discover objects from raw images. In SPACE, an image $\mathbf{x}$ is decomposed into two latent representations: foreground $\mathbf{z}^{fg}$ and background $\mathbf{z}^{bg}$. For simplicity, we only introduce the inference model of the foreground latent $\mathbf{z}^{fg}$, which is a set of object representations. We encourage the readers to refer to [20] for more details. To obtain $\mathbf{z}^{fg}$, $\mathbf{x}$ is treated as if it were divided into $H \times W$ cells and each cell is tasked with modelling at most one (nearby) object. Therefore, $\mathbf{z}^{fg}$ consists of a set of $H \times W$ object representations (i.e. $\mathbf{z}^{fg} = \{\mathbf{z}_{ij}^{fg}\}_{i=1}^{H}{}_{j=1}^{W}$), each of which is a 3-tuple[2] $\mathbf{z}_{ij}^{fg} = (z_{ij}^{pres}, \mathbf{z}_{ij}^{where}, \mathbf{z}_{ij}^{what})$. $z_{ij}^{pres} \in \{0,1\}$ is a 1-D variable that indicates the presence of any object in cell $(i,j)$, $\mathbf{z}_{ij}^{where}$ encodes $\mathbf{z}_{ij}^{fg}$'s size and location, and $\mathbf{z}_{ij}^{what}$ is a latent vector that identifies $\mathbf{z}_{ij}^{fg}$ itself. The whole inference model of $\mathbf{z}^{fg}$ can be written as:

$$q(\mathbf{z}^{fg}|\mathbf{x}) = \prod_{i=1}^{H}\prod_{j=1}^{W} q(z_{ij}^{pres}|\mathbf{x})(q(\mathbf{z}_{ij}^{where}|\mathbf{x})q(\mathbf{x}_{ij}|\mathbf{z}_{ij}^{where},\mathbf{x})q(\mathbf{z}_{ij}^{what}|\mathbf{x}_{ij}))^{z_{ij}^{pres}}, \qquad (1)$$

where $\mathbf{x}_{ij}$ is a small patch from the $\mathbf{x}$ that is obtained by the proposal bounding box identified by $\mathbf{z}_{ij}^{where}$, which should contain exactly one object if $z_{ij}^{pres} = 1$.

**Unsupervised Clustering**    After applying SPACE on the randomly collected dataset, we can obtain a set of object representations: $D = \{z_{nij}^{pres}, \mathbf{z}_{nij}^{where}, \mathbf{z}_{nij}^{what}, \mathbf{x}_{nij}\}_{n=1}^{N}{}_{i=1}^{H}{}_{j=1}^{W}$, where $N$ is the size of the dataset, $z_{nij}^{pres}, \mathbf{z}_{nij}^{where}, \mathbf{z}_{nij}^{what}, \mathbf{x}_{nij}$ are defined in Eq.(1). We first select object patches from $D$ that is of high object-presence probability by a threshold $\tau$: $D_\tau = \{\mathbf{x}_{nij} \in D : p(z_{nij}^{pres} = 1) > \tau\}$. Roughly speaking, each patch $\mathbf{x}_{nij} \in D_\tau$ should contain exactly one object. Therefore, we first project $\mathbf{x}_{nij} \in D_\tau$ into a low-dimensional latent space $\mathbf{R}^d$ by running IncrementalPCA [26] on $D_\tau$, and then adopt KMeans clustering with a given cluster number $C$ upon the projected latent representations. The object-category predictor $q(\mathbf{z}_{ij}^{cat}|\mathbf{x}_{ij}) = \Psi_{\texttt{KMeans}} \circ \Psi_{\texttt{IncrementalPCA}}$ is the composition of IncrementalPCA and KMeans:

$$\mathbf{z}_{ij}^{cat} = \Psi_{\texttt{KMeans}} \circ \Psi_{\texttt{IcreamentalPCA}}(\mathbf{x}_{ij}) \qquad (2)$$

where $\mathbf{z}_{ij}^{cat} \in \mathbf{R}^C$ is an one-hot latent which indicates the category of $\mathbf{z}_{ij}$.

For simplicity, we regard the background (i.e. $z_{ij}^{pres} = 0$) as special 'object', thus $\mathbf{z}_{ij}^{cat}$ can be written as $\hat{\mathbf{z}}_{ij}^{cat} = \Psi_{\texttt{concatenate}}([1 - z_{ij}^{pres}, z_{ij}^{pres} \cdot \mathbf{z}_{ij}^{cat}])$. Note that $\hat{\mathbf{z}}_{ij}^{cat}$ is an one-hot latent $\in \mathbf{R}^{C+1}$.

Although the object-category predictor Eq.(2) is very simple, we find it works well in our experiments. In practice, the cluster number $C$ of KMeans can be set by leveraging prior knowledge about ground-truth category number, or by analysing some clustering quality metrics such as silhouette coefficients. In more complex environments, we can also rely on other advanced unsupervised clustering methods such as NVISA [19], CC [18], etc.

### 3.2 OCAP: Object-Category Aware Perception

The OCAP module is designed to robustly incorporate the object knowledge of the (category-aware) UOD model defined in Section 3.1. In a word, OCAP is a plain convolution encoder that accepts the

---

[2]we omit $\mathbf{z}_{ij}^{depth}$ for simplicity, because we only consider 2-D environment without object occlusions in this paper.

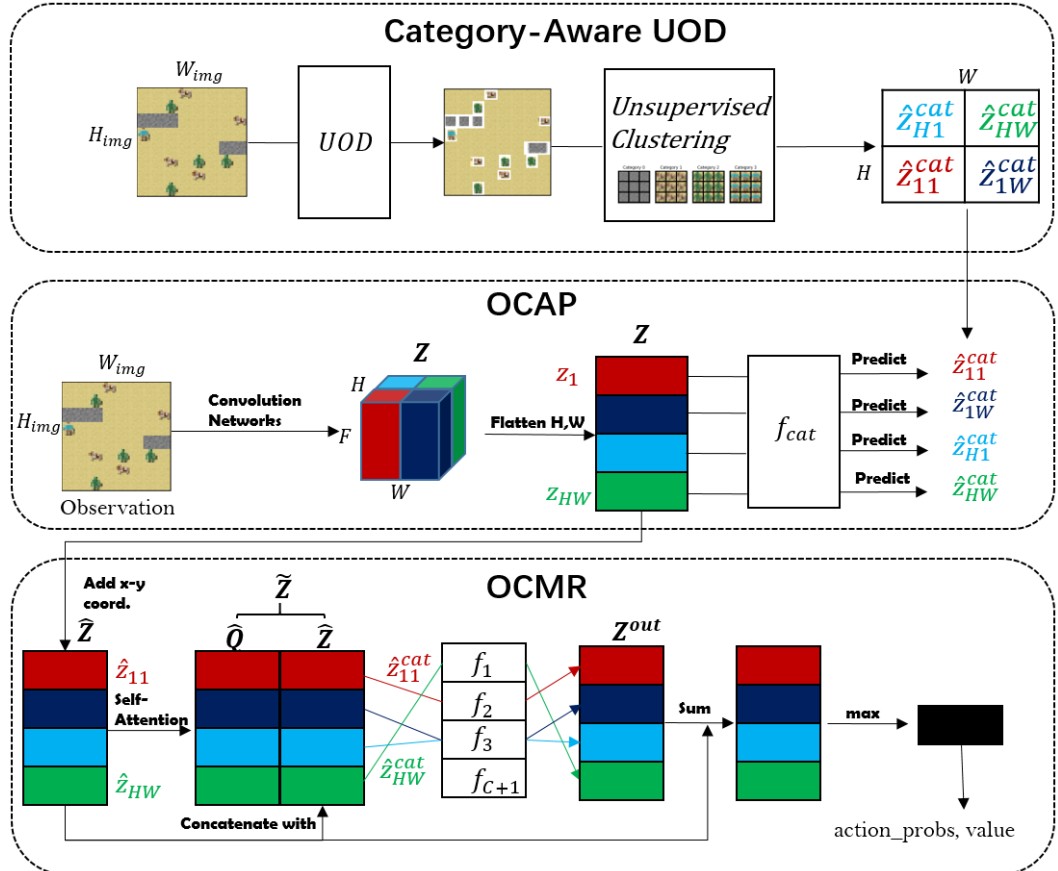

Figure 1: OCARL consists of 3 parts: (1) category-aware UOD, (2) OCAP, and (3) OCMR. (1) In category-aware UOD, the objects are firstly discovered by a UOD method (SPACE) and then assigned an (extended) category label by unsupervised clustering (IcreamentalPCA + KMeans). (2) In OCAP, the observation is first encoded into a set of features $\mathbf{Z}$ by convolution networks. Each vector in $\mathbf{z}_i = \mathbf{Z}_{:,i,j} \in \mathbf{Z}$ is given a task by the OCAP module to predict the corresponding (extended) object category, which is given by (1). $\mathbf{Z}$ is then fed to the OCMR module. (3) In OCMR, x-y coordinate information is added into $\mathbf{Z}$, which gives us $\widehat{\mathbf{Z}}$. Then we apply self-attention on $\widehat{\mathbf{Z}}$ to query related information $\widehat{\mathbf{Q}}$ for each object in $\widehat{\mathbf{Z}}$. $\widehat{\mathbf{Q}}$ and $\widehat{\mathbf{Z}}$ are concatenated to form $\widetilde{\mathbf{Z}}$. Each entity in $\widetilde{\mathbf{Z}}$ will be passed through different independent neural networks according to its corresponding object category, which will give us $\mathbf{Z}^{out}$. Finally, we perform a max operation along the $H, W$ dimension of $\mathbf{Z}^{out} + \widehat{\mathbf{Z}}$, and use the resulting feature vector to predict the value function and action probabilities.

category information from the UOD model as additional supervision signals. Such a design makes OCARL robust to the incompleteness of the UOD model: in extreme cases in which the UOD model fails to discover any objects, OCAP degenerates into a plain convolution encoder which is still able to extract useful information from the raw images with the help of reward signals.

Suppose the convolution encoder in OCAP maps the raw image observation $\mathbf{X} \in \mathbf{R}^{3 \times H_{img} \times W_{img}}$ to a latent representation $\mathbf{Z} \in \mathbf{R}^{F \times H \times W}$, where $H, W$ are the same with those in the UOD model (see Eq.(1)). On the other hand, $\mathbf{X}$ is also fed into the UOD model to get the (extended) object category information $\hat{\mathbf{z}}^{cat} = \{\hat{\mathbf{z}}^{cat}_{ij}\}^H_{i=1}{}^W_{j=1}$ via Eq.(2).

The OCAP module forces the latent representation $\mathbf{Z}$ to encode the object category information, which is implemented by training a additional category predictor $f_{cat} : \mathbf{R}^F \to \texttt{Categorical}(C+1)$. $f_{cat}$ predicts $\hat{\mathbf{z}}^{cat}_{ij}$ given $\mathbf{Z}_{:,i,j}$ which is a single channel in $\mathbf{Z}$. Therefore, the additional supervision signal that incorporates the object knowledge of the UOD model is given as:

$$L_{cat} = \sum_{i=1}^{H} \sum_{j=1}^{W} \texttt{CrossEntropyLoss}(f_{cat}(\mathbf{Z}_{:,i,j}); \hat{\mathbf{z}}^{cat}_{ij}).$$

$L_{cat}$ is used for training the convolution encoder in OCAP. In practice, $L_{cat}$ can be used as an *auxiliary loss* to any RL algorithm:

$$L_{total} = L_{RL} + \lambda_{cat} L_{cat}, \tag{3}$$

where $\lambda_{cat} \in \mathbf{R}^+$ is a coefficient.

### 3.3 OCMR: Object-Centric Modular Reasoning

OCMR is a module that takes $\mathbf{Z} \in \mathbf{R}^{F \times H \times W}$ from the OCAP module as input and outputs a feature vector that summarizes $\mathbf{Z}$. The key design philosophy of OCMR is to adopt multiple independent and object-category-specific networks, each of which focuses on processing the object features of the same corresponding category. Compared with using a universal category-agnostic network, the processing logic of each independent network in OCMR is much simpler and therefore easier to master, which in turn allows for improved generalization as we will show in Section 4.3.2. This design philosophy also agrees with the recent discovery from [22; 4] which says that neural modules of specialization can lead to better generalization ability.

Given $\mathbf{Z} \in \mathbf{R}^{F \times H \times W}$ from OCAP, we first encode the x-y coordinates information (corresponding to the $H \times W$ grid) to each channel, and then map the resulting tensor into $\widehat{\mathbf{Z}} \in \mathbf{R}^{HW \times F}$. $\widehat{\mathbf{Z}}$ is treated as $HW$ objects with x-y coordinate information encoded. To model the relations between objects, we apply a self-attention [29] module on $\widehat{\mathbf{Z}}$ to query information for each object from its related objects:

$$\widehat{\mathbf{Q}} = \texttt{softmax}(\widehat{\mathbf{Z}}\mathbf{W}_q(\widehat{\mathbf{Z}}\mathbf{W}_k)^T)\widehat{\mathbf{Z}}\mathbf{W}_v \in \mathbf{R}^{HW \times F}, \tag{4}$$

where $\mathbf{W}_q, \mathbf{W}_k, \mathbf{W}_v \in \mathbf{R}^{F \times F}$ are trainable parameters. Modeling the relations is important, because one object's high-level semantic feature can be derived from other objects. For example, whether a door can be opened is determined by the existence of the key; therefore, the door should query other objects to check whether it is openable.

After the attention module, $\widehat{\mathbf{Q}}$ and $\widehat{\mathbf{Z}}$ are concatenated together to get $\widetilde{\mathbf{Z}} = \Psi_{\texttt{concatenate}}([\widehat{\mathbf{Q}}, \widehat{\mathbf{Z}}]) \in \mathbf{R}^{HW \times 2F}$. Each $\widetilde{\mathbf{Z}}_{i,j,:}$ in $\widetilde{\mathbf{Z}}$ is then fed into *different independent neural networks* according to its corresponding object category $\hat{\mathbf{z}}_{ij}^{cat}$. In practice, this can be implemented as:

$$\mathbf{Z}_{i,j,:}^{out} = \sum_{c=1}^{C+1} f_c(\widetilde{\mathbf{Z}}_{i,j,:}) \cdot \hat{z}_{ij;c}^{cat}, \tag{5}$$

where $f_c : \mathbf{R}^{2F} \to \mathbf{R}^F$, $[\hat{z}_{ij;1}^{cat}, ..., \hat{z}_{ij;C+1}^{cat}] = \hat{\mathbf{z}}_{ij}^{cat}$. Eq.(5) can be computed in parallel, resulting a tensor $\mathbf{Z}^{out} \in \mathbf{R}^{HW \times F}$.

Finally, the $\mathbf{Z}^{out}$ is added to $\widehat{\mathbf{Z}}$ (i.e. a residual connection). We perform a max operation along the $H, W$ dimension, and using the resulting vector $\in \mathbf{R}^F$ to predict the value function and action probabilities:

$$\texttt{action\_probs, value} = f_{ac}(\max_{H,W}(\mathbf{Z}^{out} + \widehat{\mathbf{Z}})). \tag{6}$$

## 4 Experiment

### 4.1 Task Description

In this work, we consider two task domains: `Crafter` and `Hunter`. The observations on both tasks are raw images of shape $64 \times 64$.

**Crafter** [13] is a complex 2-D MineCraft-style RL task, where complex behaviors are necessary for the agent' survival. The environment is procedurally generated by arranging various resources, terrain types, and objects (18 in total). The agent is rewarded if it can craft new items and accomplish achievements (22 possible achievements in total).

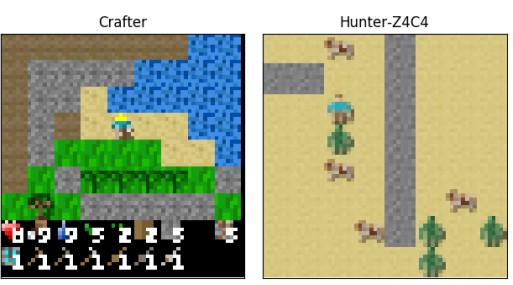

Figure 2: The `Crafter` (left) and `Hunter` (right) environment.

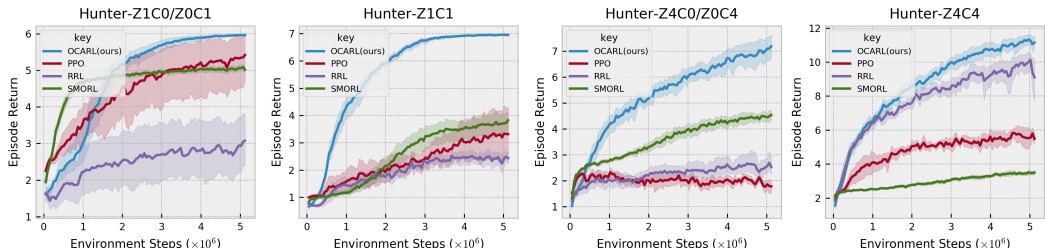

Figure 3: The results on `Hunter`. The mean returns over 12 seeds are plotted with 95% confidence interval. OCARL achieves better performance on *all* tasks; RRL is better than other baselines on `Hunter-Z4C4/Z4C4`, whereas SMORL is better on `Hunter-Z1C0/Z0C1`, `Hunter-Z1C1` and `Hunter-Z4C0`.

| Alg. | Return | Score |
|---|---|---|
| RRL | $3.58 \pm 0.80$ | $4.22 \pm 1.24$ |
| SMORL | $3.48 \pm 0.26$ | $3.94 \pm 0.50$ |
| OCARL(ours) | $\mathbf{8.14 \pm 0.35}$ | $\mathbf{12.31 \pm 0.99}$ |

Table 1: The results (averaged across 12 seeds) on `Crafter` after training for 25M environment steps. The 'Score' is a metric proposed in [13] which takes the difficultiy of each achievement into account, thus is more proper than 'Return' to benchmark agent's behaviours.

**Hunter** is much simpler than `Crafter`. It is also procedurally generated but only contains 4 types of objects: `Hunter`, `Cow`, `Zombie` and `Wall` [3]. The agent can control `Hunter` and can get positive reward (=1) if `Hunter` catches a `Cow` or kill a `Zombie`. The agent will be given a high reward (=5) if it can accomplish this task by catching/killing all `Cows` and `Zombies`. However, once `Hunter` is caught by a `Zombie`, the agent will receive a negative reward (=-1), and the episode ends. In a word, the agent in `Hunter` should master two different behavior patterns: chase & catch the `Cow`, and avoid & shoot at `Zombie`. Although `Hunter` is simple, it is a typical OORL task. We can derive different environment instances from `Hunter` by setting the number of `Zombies` and `Cows`. We use `Hunter-ZmCn` to denote an environment that spawns (m `Zombies` + n `Cows`) at the beginning of each episode, and `Hunter-ZmCn/ZnCm` an environment that spawns (m `Zombies` + n `Cows`) *or* (n `Zombies` + m `Cows`).

## 4.2 Evaluation

In this section, we evaluate OCARL's effect both on sample efficiency and generalization ability. We consider the following baselines: PPO [27], RRL [39], and SMORL [38]. PPO is a general-purpose on-policy RL algorithm with a plain neural network (i.e. Convolution + MLP). RRL proposes an attention-based neural network to introduce relational inductive biases and iterated relational reasoning into RL agent, which aims to solve object-oriented RL tasks without any other external supervision signals. SMORL utilizes the UOD model to decompose the observation into a set of object representations which are directly used as the agent's new observation. All algorithms (RRL, SMORL, and OCARL) are (re-)implemented upon PPO to ensure a fair comparison. Note that OCARL is orthogonal to the backbone RL algorithm, therefore other advanced RL methods such as [15; 11] are also applicable. For more implementation details, please refer to the Appendix.

### 4.2.1 Sample Efficiency

The results on `Hunter` domain are shown in Figure 3. We can conclude three facts from this figure: (1) OCARL performs better and is more stable than other baselines on *all* tasks. (2) RRL is competitive with OCARL on `Hunter-Z4C4` but significantly worse on other simper environments. This is because RRL relies solely on the reward signals to ground the visual images into objects; therefore, it requires enough (rewarded) interactions between objects, which simpler environments such as `Hunter-Z1C1` fail to supply. (3) SMORL performs better than other baselines (PPO, RRL) on simple environments such as `Hunter-Z1C0/Z0C1`, `Hunter-Z1C1` and `Hunter-Z4C0/Z0C4`, but unable to make progress in `Hunter-Z4C4` which consists of more objects. This is because SMORL adopts a simple reasoning module that may not be well-suited to multi-object reasoning.

---

[3]The images to render these objects come from `Crafter` [13], which is under MIT license.

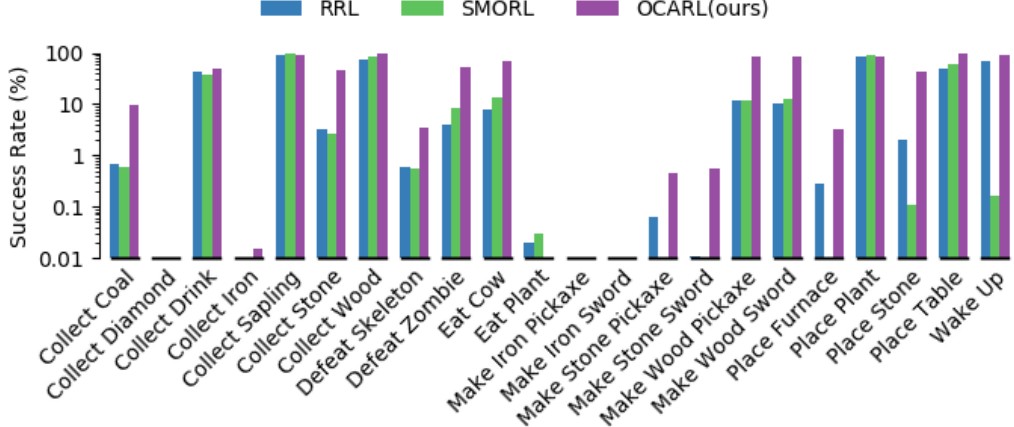

Figure 4: Success rates (averaged across 12 seeds) on 22 achievements in `Crafter`. These success rates are calculated using the final 1M environment steps during training. OCARL presents more meaningful behaviours than other baselines, such as collecting coal, defeating zombies, making wood pickaxes and so on.

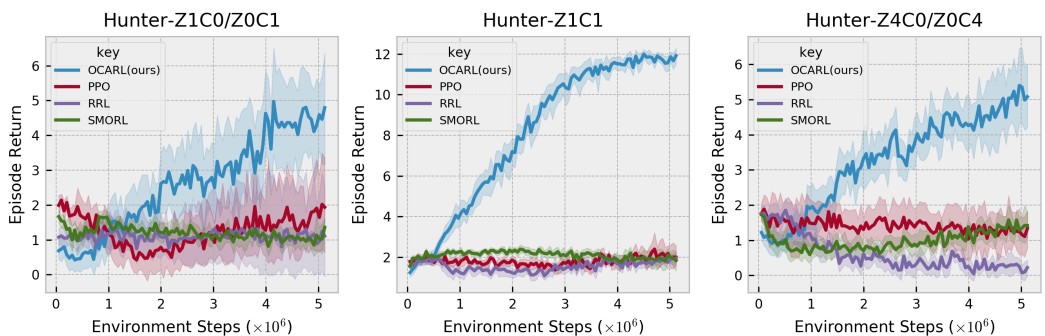

Figure 5: The generalization performance on `Hunter`. Agents are trained on different environments `Hunter-Z1C0/Z0C1`, `Hunter-Z1C1`, and `Hunter-Z4C0/Z0C4` respectively, but tested on `Hunter-Z4C4`. OCARL is the only algorithm that is able to generalize to the test environment. The mean returns over 12 seeds are plotted with 95% confidence interval.

The results on `Crafter` domain are shown in Table 1 and Figure 4. `Crafter` is a complex environment in that it features sparse (rewarded) interactions between objects and large object category numbers. Besides, some objects are omitted by the UOD model (see Appendix) because they almost do not appear in the training dataset due to the insufficient exploration of the random policy. Due to these features, both RRL and SMORL fail to provide noteworthy results (with scores of 4.22 and 3.94, respectively). On the other hand, OCARL is able to present more meaningful behaviours (with a score of 12.31), such as collecting coal, defeating zombies, making wood pickaxes and so on, as shown in Figure 4. The learning curves and detailed achievement success rates are provided in the Appendix.

### 4.2.2 Generalization

To evaluate the generalization ability, we train agents on `Hunter-Z1C0/Z0C1`, `Hunter-Z1C1` and `Hunter-Z4C0/Z0C4` separately, and then observe their test performance on `Hunter-Z4C4`. In `Hunter-Z1C1`, the agent needs to generalize from a few objects to more objects. In `Hunter-Z4C0/Z0C4`, the agent never observes the coincidence of `Cow` and `Zombie`. The generalization from `Hunter-Z1C0/Z0C1` to `Hunter-Z4C4` is the most difficult. Such a paradigm actually follows the *out-of-distribution* (OOD) setting, in that the object combination (4 `Zombies` + 4 `Cows`) is never seen in the training environments. Therefore, although such generalization is possible for humans, it is much harder for RL algorithms.

As shown in Figure 5, OCARL is the only algorithm that make significant progress on the test environment. Although other baselines can improve their training performance (Figure 3), they are

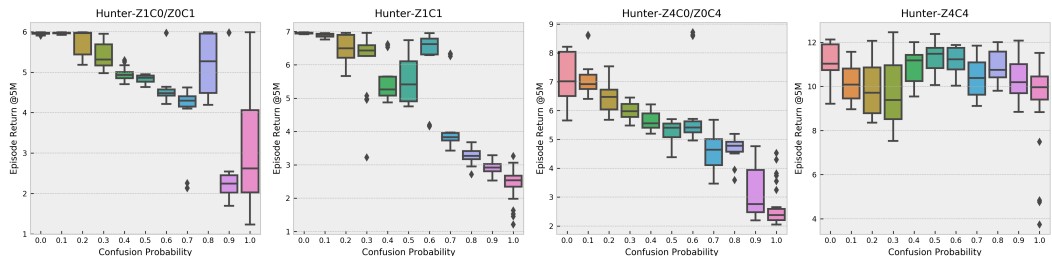

Figure 6: The Robustness of OCARL to the incompleteness of UOD methods. We randomly mask out the objects discovered by the UOD method with different confusion probability $p$ (x-axis) and then record the final performance (y-axis) over 6 seeds after training for 5M environment steps. The performance of OCARL declines as $p$ goes larger, but is still better than RRL (which exactly equals OCARL with $p = 1$)
.

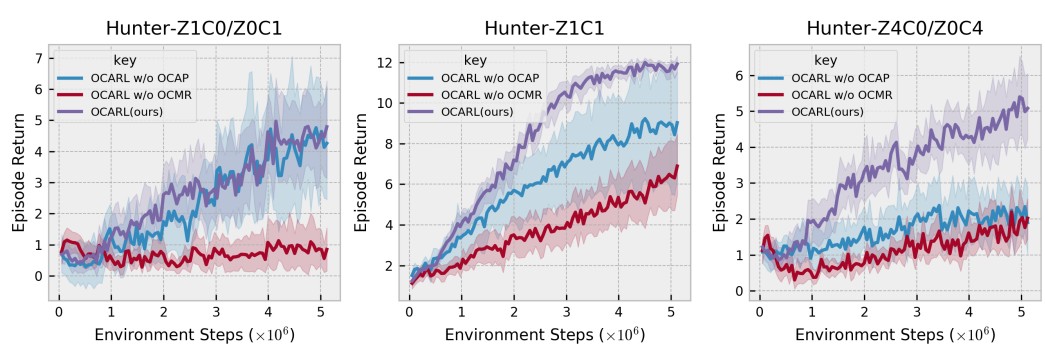

Figure 7: The generalization performance of OCARL, OCARL w/o OCMR, and OCARL w/o OCAP. The mean returns over 12 seeds are plotted with a 95% confidence interval. Without the help of OCMR, the generalization ability of OCARL will decline significantly. On the other hand, OCMR alone can bring benefits without the help of OCAP (although not as consistently as OCMR+OCAP).

unable to generalize to the test environment. The generalization performance of OCARL is most impressive on `Hunter-Z1C1`, because agent trained on `Hunter-Z1C1` even achieves slightly better than that on `Hunter-Z4C4` (i.e. directly trained on the test environment, see the last figure in Figure 3.).

### 4.3    Ablation study

#### 4.3.1    Robustness to the Incompleteness of Object Discovery Methods

OCARL relies on UOD methods to discover different objects from given observations, which are not guaranteed to discover *all* objects, especially in complex environments [34]. Therefore, one may naturally ask to what extent OCARL's performance is affected by such incompleteness.

To answer this question, we randomly mask out each object (by setting $z_{ij}^{pres}$ to 0) discovered by the UOD model with a given probability $p \in [0, 1]$, and evaluate OCARL's performance after training 5M environment steps on `Hunter`. Such random confusion of the UOD model will affect both the OCAP and OCMR modules of OCARL. Note that OCARL($p = 0$) is OCARL itself, and OCARL($p = 1$) is exactly RRL [4]. The results are shown in Figure 6. Although increasing $p$ does hinder OCARL's performance, OCARL can still benefit from the imperfect UOD model in that its performance is still better than RRL in general. Such robustness should be attributed to the fact that OCARL does not directly use the output of the UOD model as observation but instead treats it as an additional supervision signal that help the encoder to capture object-category information, which gives OCARL a second chance to find other useful information from the raw image.

| Test Environment | No Modification | Disable $f_Z$ | Disable $f_C$ |
|---|---|---|---|
| `Hunter-Z4C0` | $8.27 \pm 1.77$ | $1.17 \pm 1.16$ | $8.32 \pm 1.46$ |
| `Hunter-Z0C4` | $8.29 \pm 1.41$ | $8.13 \pm 1.78$ | $4.90 \pm 1.77$ |

Table 2: The Modularity of OCMR. We first train agent on `Hunter-Z4C4` and then disable the module $f_Z$ (corresponding to `Zombie`) or $f_C$ (corresponding to `Cow`) in OCMR (see Eq.(5)). Each agent is evaluated on both `Hunter-Z4C0` and `Hunter-Z0C4`, and the mean and standard deviation of 12 seeds are reported . The results show that the behaviour patterns (chase & catch the `Cow`) and (avoid & shoot at `Zombie`) are encoded in $f_C$ and $f_Z$ respectively.

### 4.3.2 OCMR Improves Generalization

One key advantage of OCARL is that it can generalize to unseen combinations of objects by leveraging the invariant relations between objects learnt during training. We argue that such an advantage is established via OCMR, which adopts object-category-specific networks to deal with different objects.

To confirm our argument, we consider a variant of OCARL (i.e. OCARL w/o OCMR) in which the OCMR module is removed and replaced by an RRL-like reasoning module which adopts a universal network (instead of multiple object-category-specific networks as OCARL does). The comparison between OCARL w/ and w/o OCMR is shown in Figure 7. OCARL w/o OCMR (almost completely) fails to generalize from `Hunter-Z1C0/Z0C1`, `Hunter-Z4C0/Z0C4` to `Hunter-Z4C4`, while OCARL w/ OCMR does. In Figure 7, we also consider a variant of OCARL (i.e. OCARL w/o OCAP) in which the OCAP module is disabled by setting $\lambda_{cat} = 0$ in Eq.(3). In this paradigm, only OCMR is in effect. As shown in Figure 7, OCMR alone can still bring benefits without the help of OCAP (although not as consistently as OCMR+OCAP ). According to these results, we can conclude that OCMR is crucial in OCARL's success.

### 4.3.3 The Modularity of OCMR

OCMR adopts different modules to deal with object features from different categories. In this section, we will show that each module in OCMR actually encodes an object-category-specific behaviour pattern.

As stated in Section 4.1, the agent should master two useful behaviour patterns in `Hunter`: (1) chase & catch the `Cow`, and (2) avoid & shoot at `Zombie`. The existence of these patterns can be observed by evaluating the agent in `Hunter-Z0C4` and `Hunter-Z4C0` respectively, because agents that master pattern (1) (pattern (2)) should achieve better performance on `Hunter-Z0C4` (`Hunter-Z4C0`). In Table 2, we first train agent on `Hunter-Z4C4` and then disable the module $f_Z$ (corresponding to `Zombie`) or $f_C$ (corresponding to `Cow`) by replacing $f_C$ or $f_E$ with $f_{bg}$ (corresponding to the 'background') in Eq.(5). The results shows that disabling $f_Z$ does not affect the performance on `Hunter-Z0C4` ($8.29 \rightarrow 8.13$), which means that $f_Z$ is not relative to behaviour pattern (1). Instead, $f_Z$ is corresponding to behaviour pattern (2) because we can observe a significant performance decline ($8.27 \rightarrow 1.17$) on `Hunter-Z4C0` when we disable $f_Z$. Similar phenomenon can also be observed on $f_C$.

## 5 Conclusion

In this paper, we propose OCARL, which utilize the category information of objects to facilitate both perception. For the perception, we propose the OCAP, which enables the encoder to distinguish the categories of objects. For the reasoning, we propose the OCMR, which adopts object-category-specific networks to deal with different objects. Our experiments are carried out on `Crafter` and `Hunter`. Experiments show that OCARL outperforms other baselines both on sample efficiency and generalization ability. We also perform several ablation studies to show that (1) OCARL is robust to the incompleteness of UOD methods, (2) OCMR is critical to improving generalization, and (3) each module in OCMR actually encodes an object-category-specific behaviour pattern in `Hunter`.

**Limitation** The main limitation is that we only test OCARL's generalization ability to unseen object combinations, but not novel object instances. In order to generalize to novel object instances that looks very different from old ones, the agent should interact with the novel objects first and then

---

[4]This is because Eq.(3) takes no effect and Eq.(5) also degenerates into a single universal network because all object features are passed through the same network $f_{bg}$ which is corresponding to the 'background'.

infer the their underlying categories according to these interactions. Such a problem is likely to be in the meta-RL domain and needs more efforts to solve, which we would like to leave for future work.

## Acknowledgements

This work is partially supported by the National Key Research and Development Program of China(under Grant 2018AAA0103300), the NSF of China(under Grants 61925208, 62102399, 62002338, U19B2019, 61906179, 61732020), Strategic Priority Research Program of Chinese Academy of Science (XDB32050200), Beijing Academy of Artificial Intelligence (BAAI) and Beijing Nova Program of Science and Technology (Z191100001119093) , CAS Project for Young Scientists in Basic Research(YSBR-029), Youth Innovation Promotion Association CAS and Xplore Prize.

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
