# A Implementation Details

The implementation of OCARL is available at `https://anonymous.4open.science/r/OCARL-51BF`.

**Hyper-parameters for SPACE**  In this paper, we use SPACE for object discovery. Most hyper-parameters are the same as the default parameters for Atari in SPACE, except that (1) we use a smaller network to deal with $64 \times 64$ images; (2) For `Hunter`, we disable the background module in SPACE because the observations in `Hunter` is quite simple. (3) We set the dimension of object representations $\mathbf{z}_{ij}^{what}$ to be 16, and (in `Crafter`) $\mathbf{z}^{fg}$ to be 8. The training data for SPACE is obtained via running a random policy on `Hunter-Z4C4` and `Crafter` for 100000 environment steps.

**Hyper-parameters for PPO**  Our PPO implementation is based on Tianshou [35] which is purely based on PyTorch. We adopt the default hyper-parameters in Tianshou, which are shown in Table 3.

**Hyper-parameters for OCARL**  OCARL only introduces 2 hyper-parameter $\lambda_{cat}$ (the coefficient for $L_{cat}$ defined in Eq.(3.2)) and $C$ (the number of object categories). We set $\lambda_{cat} = 0.01$ for all experiments, $C = 4, 8$ for `Hunter` and `Crafter` respectively.

**Network Architecture**  In OCAP, we use the convolution encoder from IMPALA [7], which consists of 3 residual blocks with 16, 32, and 32 channels, respectively. By the encoder, the $3 \times 64 \times 64$ image observation is encoded into $32 \times 8 \times 8$. The category predictor $f_{cat}$ is a simple 2-layer MLP with hidden size 32. The object-category-specific networks $f_1, ..., f_{C+1}$ in the OCMR module consists of $C + 1$ MLPs, each of which is of 2-layer with hidden size 64.

**Implementation for RRL**  The main implementation differences between RRL and OCARL are that (1) RRL does not use the supervision signals from the UOD model (i.e. $\lambda_{cat} = 0$ in Eq.(3)); (2) RRL adopts a different (although very similar) network architecture. In RRL, we use the same convolution encoder as OCARL. The reasoning module in RRL is similar to OCARL, except that it utilizes a non-linear universe transformation instead of multiple object-category-specific networks. We also use only one relational block in RRL as OCARL does, because we found more blocks may hinder the performance in our early experiments.

**Implementation for SMORL**  We use SPACE [20] instead of SCALOR [16] (as suggested in the SMORL paper) to obtain object representations for SMORL. Since SMORL is originally designed for goal-based RL, it uses an attention model to gather information from the set of object representations, in which the goal serves as a query vector. In our settings, we do not consider goal-based RL; therefore, we use $L$ learnable vectors as queries, which are also used in the original SMORL. We search over $L = [1, 2, 4, 8]$ and find that $L = 4$ works best in our experiments. We also find that it is better to apply an MLP to the object representations before they are fed into the attention module. To be more specific, we first run SPACE to get the object representations, sharing the same procedure as OCARL. The information of these objects is first processed by a 2-layer MLP with hidden size 32 and then gathered through an attention module in which the query consists of $L = 4$ learnable vectors. The resulting tensor (of shape $L \times 32$) is flattened and then fed into a 2-layer MLP with hidden size 64, which finally output the value function and action probabilities.

# B Details of SPACE

OCARL utilizes SPACE to discovery objects from raw objects. In Section 3.1, we have introduced the inference model of $\mathbf{z}^{fg}$. For completeness, we would like to introduce the remaining parts of SPACE in this section: (1) inference model of $\mathbf{z}^{bg}$; (2) generative model of the image $\mathbf{x}$; and (3) the ELBO to train the model.

**Inference model of $\mathbf{z}^{bg}$**  $\mathbf{z}^{bg}$ consists of several components $\mathbf{z}_k^{bg}$, and these components are inferred from the image $\mathbf{x}$ in an iterative manner: $q(\mathbf{z}_k^{bg}|\mathbf{x}) = \prod_{k=1}^{K} q(\mathbf{z}_k^{bg}|\mathbf{z}_{<k}^{bg}, \mathbf{x})$. Each component $\mathbf{z}_k^{bg}$ is futher divided into two parts: $\mathbf{z}_k^{bg} = (\mathbf{z}_k^m, \mathbf{z}_k^c)$, where $\mathbf{z}_k^m$ models the mixing weight assigned to the background component (see $\pi_k$ in the generative model of $\mathbf{x}$), and $\mathbf{z}_k^c$ models the RGB distribution ($p(\mathbf{x}|\mathbf{z}_k^{bg})$) of the background component.

| Hyper-parameter | Value |
|---|---|
| Discount factor | 0.99 |
| Lambda for GAE | 0.95 |
| Epsilon clip (clip range) | 0.2 |
| Coefficient for value function loss | 0.5 |
| Normalize Advantage | True |
| Learning rate | 5e-4 |
| Optimizer | Adam |
| Max gradient norm | 0.5 |
| Steps per collect | 1024 |
| Repeat per collect | 3 |
| Batch size | 256 |

Table 3: PPO hyper-parameters.

**Generative model of x**  The generative model consists of two parts: $p(\mathbf{x}|\mathbf{z}^{fg})$ and $p(\mathbf{x}|\mathbf{z}^{bg})$. For $p(\mathbf{x}|\mathbf{z}^{fg})$, each $\mathbf{z}^{fg}_{ij}$ is passed through a decoder to reconstruct the image patch determined by $\mathbf{z}^{where}_{ij}$. For $p(\mathbf{x}|\mathbf{z}^{bg})$, each $\mathbf{z}^{bg}_{k}$ is decoded into a background component and all components are mixed together to get the background. Foreground and background are combined with a pixel-wise mixture model to reconstruct the original image $\mathbf{x}$. The whole generative model of $\mathbf{x}$ is:

$$p(\mathbf{x}) = \int \int p(\mathbf{x}|\mathbf{z}^{fg}, \mathbf{z}^{bg})p(\mathbf{z}^{fg})p(\mathbf{z}^{bg})d\mathbf{z}^{fg}d\mathbf{z}^{bg}$$

$$p(\mathbf{x}|\mathbf{z}^{fg}, \mathbf{z}^{bg}) = \alpha p(\mathbf{x}|\mathbf{z}^{fg}) + (1 - \alpha)\sum_{k=1}^{K} \pi_k p(\mathbf{x}|\mathbf{z}^{bg}_k), \alpha = f_\alpha(\mathbf{z}^{fg}), \pi_k = f_{\pi_k}(\mathbf{z}^{bg}_{1:k})$$

$$p(\mathbf{z}^{fg}) = \prod_{i=1}^{H}\prod_{j=1}^{W} p(z^{pres}_{ij})(p(\mathbf{z}^{where}_{ij})p(\mathbf{z}^{what}_{ij}))^{z^{pres}_{ij}}$$

$$p(\mathbf{z}^{bg}) = \prod_{k=1}^{K} p(\mathbf{z}^{c}_k|\mathbf{z}^{m}_k)p(\mathbf{z}^{m}_k|\mathbf{z}^{m}_{<k})$$

In above equations, $z^{pres}_{ij}, \mathbf{z}^{where}_{ij}, \mathbf{z}^{what}_{ij}$ have been discussed in Section 3.1, $\alpha$ is foreground mixing probability, and $\pi_k$ is the mixing weight assigned to the background component $\mathbf{z}^{bg}_k$.

**The ELBO to train the model**  SPACE is trained using the following ELBO via reparameterization tricks:

$$\mathcal{L}(\mathbf{x}) = \mathbb{E}_{q(\mathbf{z}^{fg}, \mathbf{z}^{bg}|\mathbf{x})}\left[p\left(\mathbf{x} \mid \mathbf{z}^{fg}, \mathbf{z}^{bg}\right)\right] - D_{\mathrm{KL}}\left(q\left(\mathbf{z}^{fg} \mid \mathbf{x}\right)\|p\left(\mathbf{z}^{fg}\right)\right) - D_{\mathrm{KL}}\left(q\left(\mathbf{z}^{bg} \mid \mathbf{x}\right)\|p\left(\mathbf{z}^{bg}\right)\right)$$

| Achievement | OCARL(ours) | SMORL | RRL |
|---|---|---|---|
| Collect Coal | **9.5%** | 0.7% | 0.7% |
| Collect Diamond | 0.0% | 0.0% | 0.0% |
| Collect Drink | **49.0%** | 41.6% | 42.3% |
| Collect Iron | **0.0%** | 0.0% | 0.0% |
| Collect Sapling | 89.7% | **93.4%** | **89.9%** |
| Collect Stone | **46.1%** | 2.9% | 3.2% |
| Collect Wood | **98.5%** | 85.7% | 71.7% |
| Defeat Skeleton | **3.5%** | 0.5% | 0.6% |
| Defeat Zombie | **51.4%** | 11.3% | 3.8% |
| Eat Cow | **67.2%** | 11.6% | 7.6% |
| Eat Plant | 0.0% | 0.0% | 0.0% |
| Make Iron Pickaxe | 0.0% | 0.0% | 0.0% |
| Make Iron Sword | 0.0% | 0.0% | 0.0% |
| Make Stone Pickaxe | **0.4%** | 0.0% | 0.1% |
| Make Stone Sword | **0.6%** | 0.0% | 0.0% |
| Make Wood Pickaxe | **86.1%** | 15.2% | 12.0% |
| Make Wood Sword | **83.5%** | 17.8% | 10.6% |
| Place Furnace | **3.2%** | 0.0% | 0.3% |
| Place Plant | 86.9% | **88.4%** | 83.0% |
| Place Stone | **44.1%** | 1.2% | 2.1% |
| Place Table | **95.6%** | 67.5% | 49.8% |
| Wake Up | **88.1%** | 0.5% | 70.2% |
| Score | **12.3%** | 3.9% | 4.2% |

Table 4: Success rates on 22 achievements in `Crafter`.

## C  More Results on `Crafter`

In Figure 8, we plot the learning curves on the `Crafter` domain. Only OCARL is able to make progress continuously. Both RRL and SMORL get stuck in local minimal after training for about 1M environment steps.

In Table 4, we give the detailed success rates for 22 achievements on `Crafter`, which is also reported in Figure 4. As shown in Table 4, OCARL presents more meaningful behaviours such as collecting coal, defeating zombies, making wood pickaxe and so on.

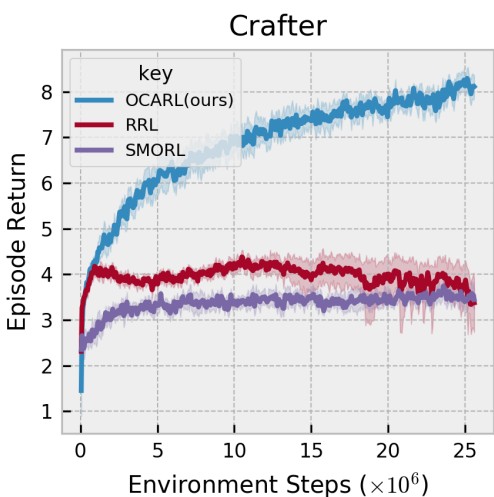

Figure 8: The learning curves on `Crafter`. The mean returns over 12 seeds are plotted with a 95% confidence interval. Both RRL and SMORL fail to make progress after the initial 5M environment steps, while OCARL does.

| water | grass | stone | path | sand | tree | lava | coal | iron |
|-------|-------|-------|------|------|------|------|------|------|
| 2.76 | 46.5 | 1.09 | 0.36 | 0.94 | 2.24 | 0.018 | 0.094 | 0.0294 |

| diamond | table | furnace | player | cow | zombie | skeleton | arrow | plant |
|---------|-------|---------|--------|------|--------|----------|-------|-------|
| 0.0025 | 0.0108 | 0 | 1 | 0.495 | 0.14 | 0.0947 | 0.0973 | 0.165 |

Table 5: The average number of objects in a single image. These numbers are obtained on the training dataset of SPACE via analysing the ground truth object category label provided by `Crafter`.

## D  Analysis of Unsupervised Object Discovery

In Figure 9, we plot the discovered object categories in `Crafter` and `Hunter`. Since `Hunter` is quite simple, our algorithm can assign correct categories for all objects. In `Crafter`, there exist 18 kinds of objects in total, making it much harder than `Hunter`. Since we use only 8 categories, there are some cases that multiple objects with different ground-truth categories are predicted into the same category (such as `Category` 5 and 7 in Figure 9). Although there exist assignment mistakes in `Crafter`, OCARL is still much better than other baselines.

Figure 9 also shows that some objects in `Crafter` is omitted by the UOD model, such as [image], [image], [image], [image], [image] and [image]. This is probably because the some objects almost does not appear in in training dataset (Table 5) or are treated as background by SPACE.

In Figure 10, we report the object discovered by SPACE. The combination of SPACE and unsupervised clustering on the discovered objects (i.e. Eq.(2)) can tell us not only where one object locates, but also what category it is.

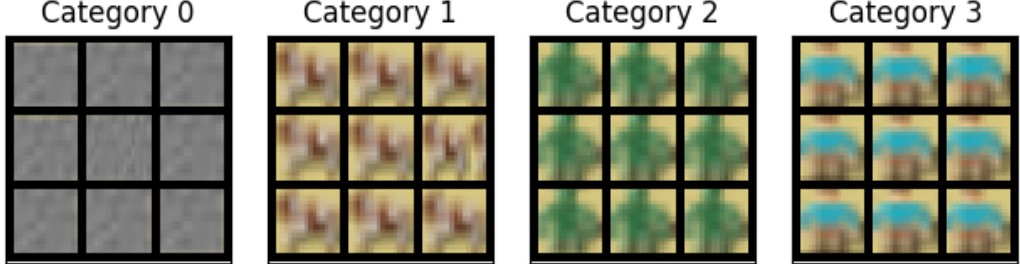

(a) The discovered object categories on `Hunter`.

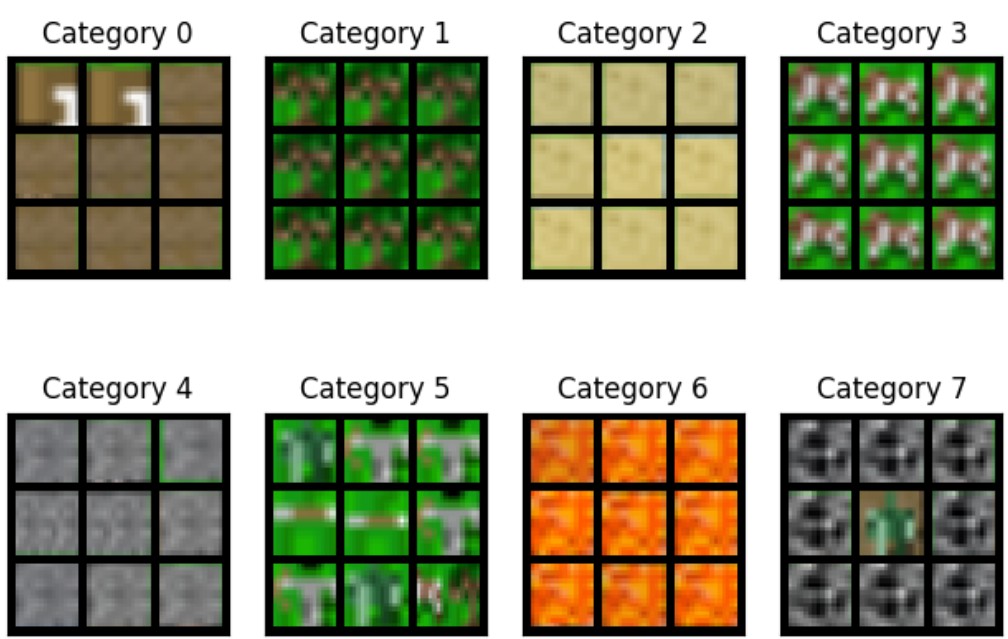

(b) The discovered object categories on on `Crafter`

Figure 9: The objects discovered by Eq.(2). In `Hunter`, object categories are perfectly predicted, whereas in `Crafter` there exist some mistakes.

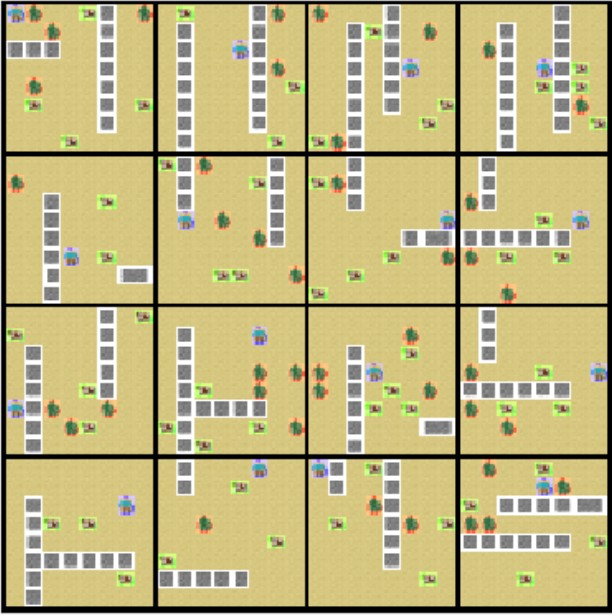

(a) The objects discovered on `Hunter`.

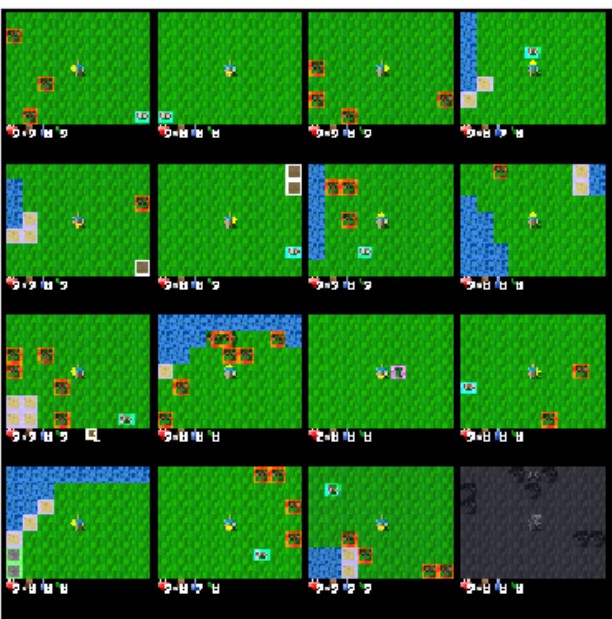

(b) The objects discovered on `Crafter`

Figure 10: The objects discovered by SPACE. The objects are bound by boxes whose colour indicates its corresponding category that is predicted by Eq.(2). In `Crafter`, the object centred in each image is ignored by the SPACE because its location never changes and thus is distinguished as background.