# OpenReview forum: "Object-Category Aware Reinforcement Learning"
_NeurIPS.cc/2022/Conference — NeurIPS 2022 Accept_

### Official Review · Reviewer_dKY1 · 2022-07-10

**Rating:** 5
**Confidence:** 3
**Soundness:** 2 fair
**Presentation:** 3 good
**Contribution:** 2 fair

**Summary:**

This paper proposed a object-oriented reinforcement learning method, wihich consists of three parts: Category-Aware UOD, OCAP and OCMR. Category-Aware UOD mainly provides category supervision information for OCAP, enabling OCAP to perceive objects in the environment more accurately．OCMR learns the interactions between objects in a self-supervised manner to better predict action probabilities. Through the recognition of object categories in the environment and the learning of interrelationships, the model can learn better action strategies. It is verified that the proposed model can better perform reinforcement learning tasks in two experimental environments.

**Questions:**

1) Why do you need to take the maximum value of Z^out and Z in Eq. 6?
2）In the OCAP module, it is surprising that different objects can be segmented unsupervised from the environment image only by a simple convolutional neural network. Can you explain why this works?

**Limitations:**

1) The proposed method does not consider the experimental environment with occlusion between objects, which is unreasonable.

**Strengths And Weaknesses:**

Strengths:
1）It is an interesting strategy to help the agent better perform reinforcement learning tasks by identifying the types of objects in the environment.
2) The experimental results show that the proposed method is effective.

---

> ### Author Response · Authors · 2022-08-01
> **Response**
>
> Thank you very much for your comments. Please feel free to communicate with us if you have further questions or suggestions.
>
> >Q1: Why do you need to take the maximum value of Z^out and Z in Eq. 6?
>
> $\max$ is actually an aggregation operator upon $Z^{out} + Z$, which is widely used in many applications. For example, in GNN, an aggregation operator (e.g., max, sum, mean,...) is often utilized to conclude the information of the whole graph.
>
> >Q2: In the OCAP module, it is surprising that different objects can be segmented unsupervised from the environment image only by a simple convolutional neural network. Can you explain why this works?
>
> This is because this CNN is supervised by object category labels that come from the UOD model. Actually, the $p(z^{fg}|x)$ in SPACE also contains a CNN that infers object information from the raw images. In our opinion, CNN is naturally suitable for segmenting the image because the receptive field of each channel in CNN is different and thus can model different areas of the original image.
>
> >Q3: The proposed method does not consider the experimental environment with occlusion between objects, which is unreasonable.
>
> In most RL tasks, there do not exist occlusion between objects. Occlusion between objects is actually a classic problem in the unsupervised object discovery domain. In general, it requires a recurrent architecture to infer the occlusion, making the whole process much more complicated. Therefore, we would like to leave this for future work.

---

### Official Review · Reviewer_1n95 · 2022-07-10

**Rating:** 4
**Confidence:** 5
**Soundness:** 2 fair
**Presentation:** 2 fair
**Contribution:** 2 fair

**Summary:**

This paper proposes a new framework applied to Object-oriented reinforcement learning (OORL) called Object-Category Aware Reinforcement Learning (OCARL), which aims to explicitly model the similarity between different objects of the same category. It consists of three main components: 1) UOD: unsupervised learning method (SPACE+IncrementalPC+KMeans) is used to complete the identification and clustering of objects; 2) OCAP: the clustering results are used as supervised learning signals to guide the encoding of objects; 3) OCMR: a self-attention mechanism is introduced, and different category objects use independent networks to complete the reasoning process. The experiments show that OCARL can improve the sample sampling efficiency and generalization ability of the model.

**Questions:**

1. Are there experiments that show it is possible to continue training the UOD in the RL phase?? It would make the results stronger and less reliant on a good random exploration strategy.
2. When the number of preset categories is greater than the actual number of categories (e.g. Hunter preset category number is greater than 4), is there an experimental display?
3. The clustering results of unsupervised learning are shown in Appendix C, but I am more interested in the representation of the latent variable $Z$. Is it possible to show the prediction results of $f_{cat}$ in OCAP?
4. Does the method only apply to environments, which can be divided into grids? In line 105, the image is divided into $H\times W$ cells, and each cell corresponds to an object. if the object is irregular or of a different size, how to handle it?

**Ethics Review Area:**

["I don’t know"]

**Limitations:**

The paper doesn't really discuss its own limitations explicitly. Adding a section would be quite helpful. OCARL requires additional information about the environment (the number of categories), which may not work well if the difference between the number of preset categories and the actual number of categories in the environment is large.

**Strengths And Weaknesses:**

Strengths：

1. The paper contributes to improving object representation learning in model-free RL which has practical applications in object-oriented RL.
2. Compared with existing methods, OCARL shows outstanding generalization ability.
3. The paper is well written and well organized.

Weaknesses：

1. The paper gives an implementation that combines unsupervised learning, supervised learning, and reinforcement learning. Each of the sections chooses a classic implementation, but the reasonableness is not well illustrated and there is no experiment showing the advantages compared with other implementable methods.
2. The paper points out that category information improves exploration efficiency and generalization, but there is no essential reason for the improvement (derivation, proof, etc.), nor is the condition and scope of its use stated.
3. The UOD method uses data derived from the random exploration, but random policies often fail to explore the environment effectively and may result in some of the objects not being discovered.
4. This paper uses KMeans for clustering operation, so the number of categories k is the hyperparameter. In practical scenarios, we usually do not know the specific number of categories of objects explicitly, so there may be cases where the preset number of categories is greater or less than the actual number of categories. This paper only implements the "less" situation in the Crafter environment without considering the case where the preset number of categories is greater than the actual number of categories.
5. Generalizability requires that the model be trained and applied to new data or new environments. However, the UOD is trained using the same environment as the test.
6. There may be a formatting error at the top of page 5, where the first two lines have no line number and the formula is not coded, resulting in an error in the formula reference in line 259.

---

> ### Author Response · Authors · 2022-08-01
> **(Part1/2)Response**
>
> Thanks for your very detailed review. We found your concerns (especially Q4) to be very constructive, which inspire us to improve our work further.
>
> > Q1: The paper gives an implementation that combines unsupervised learning, supervised learning, and reinforcement learning. Each of the sections chooses a classic implementation, but the reasonableness is not well illustrated and there is no experiment showing the advantages compared with other implementable methods.
>
> The methods consist of three parts: UOD, OCAP, OCMR. In this paper, we mainly focus on harnessing the object information provided by UOD; therefore, we think the key contribution of this paper is the proposal of OCAP and OCMR. For both OCAP and OCMR, we do describe the advantages of our design at the beginning of the corresponding sections, and these advantages are also supported by experiments in Section4.3. For UOD, it is true that we do not describe why we choose SPACE & KMeans in Section3.1 because we do not take these as our main contributions, and they can be replaced by other methods whenever possible.
>
> >Q2: the paper points out that category information improves exploration efficiency and generalization, but there is no essential reason for the improvement (derivation, proof, etc.), nor is the condition and scope of its use stated.
>
> There are several reasons that can account for the improvement:(1) By leveraging the category information, OCARL can learn to decompose the raw observations into a set of objects in an unsupervised fashion, which are disentangled representations and beneficial to the agent's understanding of the environment. The OCARL agent can focus on exploring the object functionalities without figuring out objects via reward signals, which will enhance the exploration efficiency. (2) Based on the object representations, OCARL builds a policy that makes decisions by reasoning over the functionalities and relationships of/between objects, which are invariant across different environment instances. By learning the invariant part between train and test environment, OCARL can generalize better. (3) OCARL adopts multiple independent and object-category-specific networks, each of which focuses on processing the object features of the same corresponding category. The processing logic of each independent network in OCMR is much simpler and, therefore, easier to master.
>
> >Q3: The UOD method uses data derived from the random exploration, but random policies often fail to explore the environment effectively and may result in some of the objects not being discovered.
>
> This is exactly why we design OCAP. Thanks to OCAP, OCARL is robust to the incompleteness (i.e., ignoring some objects) of the UOD model, which has been demonstrated by experiments in Section4.3.1. Actually, such phenomenon (i.e., some objects not discovered) can be observed on the Crafter, in which OCARL has achieved impressive results.
>
> >Q4: This paper uses KMeans for clustering operation, so the number of categories k is the hyperparameter. In practical scenarios, we usually do not know the specific number of categories of objects explicitly, so there may be cases where the preset number of categories is greater or less than the actual number of categories. This paper only implements the "less" situation in the Crafter environment without considering the case where the preset number of categories is greater than the actual number of categories.
>
> In our current experiment setting, the number of categories (C) is set by the oracle. However, it is easy to automatically find a proper (C). For example, we can use a clustering method (instead of KMeans) that does not need to specify (C) in advance (instead, the clustering method tells us the optimal (C)). Besides, there exist many metrics (such as Silhouette Coefficient) to measure the quality of clustering and can be utilized to find a proper (C). For example, in Hunter, we can derive Silhouette Coefficients for K=2,...,10:
>
> |K|2|3|4|5|6|7|8|9|10|
> |-|-|-|-|-|-|-|-|-|-|
> |silhouette coefficient|0.648|0.876|0.961|0.950|0.942|0.942|0.923|0.646|0.633|
>
> and automatically find that C=4 is the most proper choice. What's more, OCARL is also robust to the case when (C) is given 'more' than the ground-truth category number, as we show in this figure:
> https://anonymous.4open.science/r/NIPS2022-Rebuttal-FAD3/diff_C.png
> In this figure, we set C=6 in Hunter (who has 4 ground-truth object categories), and find that the resulting performance is almost the same as C=4. This is because over-segmented clustering results provided by KMeans also contain category information, which can be used by OCAP+OCMR.

---

> > ### Author Response · Authors · 2022-08-01
> > **(Part 2/2)Response**
> >
> > >Q5: Generalizability requires that the model be trained and applied to new data or new environments. However, the UOD is trained using the same environment as the test.
> >
> > Thanks for your advice. We have conducted an experiment in which the UOD model is trained on Hunter-Z1C0/Z0C1 but then used to capture objects on Hunter-Z4C4; here is the result: https://anonymous.4open.science/r/NIPS2022-Rebuttal-FAD3/z1c0.png
> > Here, the third image shows that the UOD model is able to correctly discover all objects and their corresponding categories (marked by different colors). The above results show that the UOD model itself is able to generalize well to novel object combinations, and therefore the results in Section4.2.2(Generalization) should not change when we use data from source tasks to train UOD.
> >
> > >Q6: There may be a formatting error at the top of page 5, where the first two lines have no line number and the formula is not coded, resulting in an error in the formula reference in line 259.
> >
> > We have corrected this issue in our latest revision.
> >
> > >Q7: Are there experiments that show it is possible to continue training the UOD in the RL phase?? It would make the results stronger and less reliant on a good random exploration strategy.
> >
> > Training the UOD online is actually a continual learning problem. In our early experiments, we found that these novel objects are always recognized as background (instead of the foreground) by the UOD model. We think more efforts in the UOD domain are needed to support training UOD online.
> >
> > >Q8: When the number of preset categories is greater than the actual number of categories (e.g. Hunter preset category number is greater than 4), is there an experimental display?
> >
> > In https://anonymous.4open.science/r/NIPS2022-Rebuttal-FAD3/diff_C.png, we set C=6 in Hunter and find the results are almost the same with (C=4). See Q4 for more analysis.
> >
> > >Q9: The clustering results of unsupervised learning are shown in Appendix C, but I am more interested in the representation of the latent variable . Is it possible to show the prediction results of $f_{cat}$  in OCAP?
> >
> > Actually Appendix C also shows the prediction results of $f_{cat}$, see the colored boxes in Figure10.
> >
> > >Q10: Does the method only apply to environments, which can be divided into grids? In line 105, the image is divided into $H\times W$ cells, and each cell corresponds to an object. if the object is irregular or of a different size, how to handle it?
> >
> > Objects of different sizes can be handled by SPACE. Line 105 means that we have $H\times W$ object slots, and each slot $(i, j)$ is tasked to capture the object $nearby$ the anchor point $(\frac{iH_{img}}{H}, \frac{jW_{img}}{W})$ in the image. This object is not necessarily of a regular size. For example, the bounding boxes (i.e., the captured objects) in
> > https://anonymous.4open.science/r/NIPS2022-Rebuttal-FAD3/space_diff_obj_size.PNG (a detection result from the paper of SPACE)
> > is actually of different shapes. In fact, $z_{ij}^{where}$ (one output of SPACE, see Eq.(1)) contains a set of parameters for a spatial transformer network that can automatically select the object patch and then resize it into a regular size.

---

### Official Review · Reviewer_2Frs · 2022-07-10

**Rating:** 7
**Confidence:** 4
**Soundness:** 4 excellent
**Presentation:** 3 good
**Contribution:** 4 excellent

**Summary:**

The paper proposes learning object categories in order to enhance object-oriented RL (OORL). While past work on OORL has shown the benefit of extracting objects from a scene before learning with RL, this work has not considered learning to cluster objects into related categories so that the policy can treat similar objects the same. This work uses an existing Unsupervised Object Discovery (UOD) method to discover objects, but then learns a clustering of the resulting objects. It then asks the RL network to not only be able to predict the category of objects (forcing the representation to learn about object categories), but it also uses the object categories to apply independent category-specific networks to each object, which the authors argue improves modularity and therefore generalization. The paper shows results in two gridworld domains which represent complex sequential decision making tasks, and show significant improvements over relevant baselines and ablations.

**Questions:**

If you can explain the difference between RRL and OCARL and clarify the issues I have raised above I may be willing to increase my score.

Why were most of the results shown in Hunter rather than Crafter? Crafter is a complex unsolved domain so it is of significant interest to the community. I would suggest moving Figure 8 from the appendix into the main text if possible, since those results are compelling.

I would suggest removing the italics such as "OCARL achieves better performance on *all* tasks" and "is the *only* method". This gives an appearance of a lack of objectivity that is not appropriate for a scientific paper.

I suggest adding a sentence after line 89 to make it clear how OCARL is distinct from SMORL.

The paper contains several English-language errors that should be proof-read and corrected. A non-exhaustive list:
- Line 23: "To deal with these limitations, OORL is a promising way" -> "OORL is a promising way to deal with these limitations"
- Line 25: invariant -> invariance
- Line 29: "yield in low generality" doesn't make sense.
- Line 158: concatenate -> concatenated

I appreciated the cognitive science justification for object awareness that the authors provided.

**Limitations:**

The authors acknowledge the limitations of their clustering approach in line 124-126, but did not otherwise acknowledge the limitations of their method. I would suggest acknowledging that the didn't test generalization to novel instances of the same object category, as I explained above.

It does not appear that the authors included a discussion of the societal impact of their work. Perhaps they can talk about the benefit of improved object-oriented RL to the potential for building better robots, and the consequences thereof.

**Strengths And Weaknesses:**

### Strengths
**Originality**: The idea of learning categories of objects, and a policy that treats objects in the same category in a similar way, is both compelling and novel. The ablation studies reveal that the paper essentially makes two novel algorithmic contributions based on object category awareness that improve performance: OCAP (predict object cluster labels), and OCMR (having different modular independent networks for different object categories).

**Significance:**
The paper benchmarks against two relevant OORL baselines (RRL and SMORL), and a reasonably advanced RL method (PPO), and shows clear and significant performance gains above these methods in complex sequential decision making tasks including Crafter (which is based on MineCraft). Although these environments are gridworlds, crafting is a complex problem and it is difficult for conventional methods to obtain high rewards. The idea of reasoning over object categories is likely to be useful in a broader range of tasks (e.g. object manipulation). The generalization performance is also interesting.

**Quality**:
The design of the network architecture proposed in the paper appears to be complex but thoughtful; the authors carefully justify each design decision (e.g. why the network should predict the object category as an objective, rather than use it as input, so it can be robust if the UOD module fails to extract the right category). The ablations also back up the claims that each component is necessary.

**Clarity**:
The description of related work is clear and gives a good overview of the field while drawing the distinction with this work.

### Weaknesses:
**Significance/Quality:**
In spite of the fact that experiments were conducted in Crafter, most of the analysis for the paper is shown in Hunter. Crafter is a much harder environment of more significance to the community, so it would have been nice to see learning curves and ablations in this environment as well.

Further, although the paper tests generalization, it does so in a pretty limited way, because in no experiment is the agent ever tested on a truly novel object (only on differing combos of zombies and cows, but it has always seen at least one zombie or cow previously). The current paper actually does not test generalization to novel instances of the same category. It would be interesting to extend the experiments to another domain (one example could be simulated robotic manipulation of objects on a tabletop), and test whether the method can truly generalize to new instances of the same category. For example, if the agent had trained to pick up a red cup and a blue cup, could it generalize to picking up a green cup?

**Clarity**:
The biggest weakness of the paper is clarity. In addition to smaller grammatical errors, there are some places where explanations significantly detract from clearly understanding the paper. The single biggest issue is in the method/results. Line 242-243 states that OCARL with p=1 is exactly RRL (where p is the probability that the UOD module can't detect a category). If this were true, this would significantly detract from the novelty of the paper. However, a few sentences below in lines 253-255 the paper directly contradicts this statement, by pointing out that OCARL uses the OCMR module, while RRL uses a universal network "instead of multiple object-category-specific networks like OCARL does". One of these two statements must be false, and I believe it is the one about the exact equivalence with RRL. This should be clarified, and it would be good to further explain in the related work or methods how the architecture being proposed is different than RRL.

The modularity experiment in Section 4.3.3 seemed interesting, but it was so hard to follow lines 272-276 I was not sure what the experiment actually showed. Is there a table or figure describing the results?

 A non-exhaustive list of other clarity issues:
- There are so many acronyms in the paper that it is hard to follow what's actually going on. Rather than OCAP and OCMR, it might be nice to use a phrase like "category modules" to cue the reader.
- In Section 3.1, it is not clear what $z_{ij}^{what}$ actually is... is it a continuous embedding? A categorical label?
- Line 106 appears to state that $z^{fg}$ consists of a fixed number (H x W) of object representations, but the related work section states that this is only a drawback of spacial mixture models, and not SPACE (which uses a spatial attention module for the foreground).
- Lines 100-101 contain an out of place detail (we run a random policy to get data and apply SPACE) that seems premature given the environment has not been introduced at this point. It is hard to evaluate whether a random policy would be able to collect enough data to cover all object categories without knowing about the environment. This also does not seem to be part of the *method*, per se, but rather an implementation detail.
- The explanation in lines 150-157 is unclear.
- Can Equation 5 be interpreted as applying attention weights based on the category probabilities predicted from Z? The explanation / rationale should be improved here.
- Is OCARL optimized with PPO as well? This should be clarified.

---

> ### Author Response · Authors · 2022-08-01
> **(Part 1/2) Response**
>
> We thank the reviewer for the insightful comments and suggestions.We would like to provide detailed explanations to your comments:
> >Q1: Crafter is a much harder environment of more significance to the community, so it would have been nice to see learning curves and ablations in this environment as well.
>
> Thanks for your advice. Although Crafter is of more significance, it does not provides an easy way to control the generating progress of environment instances (such as the spawning distribution of each object category), which is needed in our experiments (such as Section4.2.2, Section4.3.2, and  Section4.3.3). Besides, it will take much more time to run a trial in Crafter.
>
> >Q2: The current paper actually does not test generalization to novel instances of the same category
>
> Yes, in the current paper, we are more interested in the generalization to unseen object combinations. Generalization to novel object instances does make sense in many applications and actually is exactly the topic of our next research. However, we think such generalization needs more effort to achieve. Generally speaking, such generalization needs the agent to interact with the novel object instances first and then infer their underlying categories according to these interactions. Such a problem is likely to be in the meta-RL domain, and we might need to train a separate inference model to predict objects' categories given their corresponding interactions.
>
>
> >Q3: [(1) Line 242-243 states that OCARL with p=1 is exactly RRL] v.s. [(2) OCARL uses the OCMR module, while RRL uses a universal network "instead of multiple object-category-specific networks like OCARL does"].
>
> (1) and (2) are both true and compatible. The main difference between RRL and OCARL is that OCARL harnesses the object category information discovered by UOD, which is achieved by Eq.(3) and Eq.(5). When p=1, the UOD cannot discover any objects, therefore all objects are labelled into "background". In such a scenario, the OCAP degenerates into a plain convolution encoder because the UOD model cannot provide any object information (i.e., Eq.(3) has not effect). The object-category-specific networks (Eq.(5)) in OCMR also degenerate into a single universal network because all object features are passed through the same network $f_{bg}$. Therefore, OCARL with p=1 is exactly the same as RRL.
>
> >Q4: The modularity experiment in Section 4.3.3 seemed interesting, but it was so hard to follow lines 272-276
>
> This section mainly shows that $f_C, f_Z$ (two object-category-specific networks in Eq.(5)) are one-to-one corresponding to behavior patterns (P1) chase&catch the Cow and (P2) avoid & shoot at Zombie, respectively. Line 272-276 shows that once $f_Z$ is disabled, (P2) disappears while (P1) still remains. "Disable $f_Z$" is achieved by replace the parameters of $f_Z$ with parameters of $f_{bg}$. Whether (P1) (or (P2)) disappears or remains is checked by running the resulting policy with certain networks disabled on Hunter-Z0C4 (or Hunter-Z4C0), which is shown in Table2. We have corrected an error in line 274, which may account for your incomprehension.
>
> >Q5: There are so many acronyms in the paper that it is hard to follow what's actually going on.
>
> Thanks for your advice. There are four widely used acronyms in our paper: (1) OCARL; (2) UOD; (3) OCAP; (4) OCMR, and all these acronyms have been put together in Figure1 to enable easy reference. We think using these acronyms can make our paper more concise and accurate.
>
> >Q6: In Section 3.1, it is not clear what $z_{ij}^{what}$ actually is... is it a continuous embedding? A categorical label?
>
> $z_{ij}^{what}$ is a latent embedding of an object, which can be used to reconstruct the object. All categorical labels in the paper should be with superscript $\cdot^{cat}$
>
> >Q7: Line 106 appears to state that $z^{fg}$ consists of a fixed number (H x W) of object representations, but the related work section states that this is only a drawback of spacial mixture models, and not SPACE (which uses a spatial attention module for the foreground).
>
> Traditional spacial mixture model methods only allow several object slots (e.g. In MONet, slot number=7). However, SPACE has $H\times W$ object slots in total, which are sufficient for most tasks (in our experiment, it is $8\times 8=64$). Besides, we can easily increase $H,W$ to introduce more slots.
>
> >Q8: Lines 100-101 contain an out of place detail (we run a random policy to get data and apply SPACE) that seems premature given the environment has not been introduced at this point.
>
> Thanks for your advice. We have moved this sentence to the Appendix.
>
> >Q9: The explanation in lines 150-157 is unclear.
>
> Line 150-157 describes 2 processes: (1) add x-y coordinate information (2) apply a self-attention module to model the relations between objects. Eq.(4) is just a formula of self-attention, where $\hat{Z}$ is used as query, key, and value. We have rewritten these sentences to make them clearer.

---

> > ### Author Response · Authors · 2022-08-01
> > **(Part2/2) Response**
> >
> > >Q10: Can Equation 5 be interpreted as applying attention weights based on the category probabilities predicted from Z?
> >
> > Yes. Note that such attention weight is just a one-hot vector.
> >
> > >Q11: Is OCARL optimized with PPO as well? This should be clarified.
> >
> > Yes. This has been clarified in line 198.
> >
> > >Q12: If you can explain the difference between RRL and OCARL and clarify the issues I have raised above I may be willing to increase my score.
> >
> > see Q3.
> >
> > >Q13: Why were most of the results shown in Hunter rather than Crafter?
> >
> > This is because Hunter is a flexible domain in which we have full control of the generating mechanism of different environment instances, making it possible for us to test OCARL's properties (such as OOD generalization). On the other hand, Crafter does not provide an easy way to control the generating progress of environment instances and runs much slower than Hunter.
> >
> > >Q14: Other advice
> >
> > We thank you for your advice on improving the paper, and we have considered it in our latest revision.

---

> > ### Comment · Reviewer_2Frs · 2022-08-03
> > **Thanks for your response**
> >
> > > Yes, in the current paper, we are more interested in the generalization to unseen object combinations. Generalization to novel object instances does make sense in many applications and actually is exactly the topic of our next research. However, we think such generalization needs more effort to achieve. Generally speaking, such generalization needs the agent to interact with the novel object instances first and then infer their underlying categories according to these interactions. Such a problem is likely to be in the meta-RL domain, and we might need to train a separate inference model to predict objects' categories given their corresponding interactions.
> >
> > Okay, that makes sense to me.
> >
> > > (1) and (2) are both true and compatible. The main difference between RRL and OCARL is that OCARL harnesses the object category information discovered by UOD, which is achieved by Eq.(3) and Eq.(5). When p=1, the UOD cannot discover any objects, therefore all objects are labelled into "background". In such a scenario, the OCAP degenerates into a plain convolution encoder because the UOD model cannot provide any object information (i.e., Eq.(3) has not effect). The object-category-specific networks (Eq.(5)) in OCMR also degenerate into a single universal network because all object features are passed through the same network . Therefore, OCARL with p=1 is exactly the same as RRL.
> >
> > I see what you're saying now, but this is still unclear in the paper. It would be best to add this explanation to the paper (especially including "The object-category-specific networks (Eq.(5)) in OCMR also degenerate into a single universal network because all object features are passed through the same network"), or rephrase this section more clearly.

---

> > > ### Author Response · Authors · 2022-08-05
> > > **About the explanation of Q3**
> > >
> > > Thanks for your advice. We have uploaded a new revision that includes this explanation.

---

### Official Review · Reviewer_B8LM · 2022-07-12

**Rating:** 7
**Confidence:** 3
**Soundness:** 4 excellent
**Presentation:** 3 good
**Contribution:** 3 good

**Summary:**

The paper proposes augmenting Object Oriented RL approaches with additional information in the form of object category. Specifically, they use a module which uses unsupervised clustering to identify object categories from the output of a (previously known) unsupervised object detection algorithm. This is additional info (predicted category) is then incorporated using a convolutional encoder (OCAP). The object-category aware information from this module is passed into the third module which contains category-specific neural networks, which finally returns the action probabilities and values.

**Questions:**

1. Is the number of classes (C) a known, required parameter? Was this known for the games, or not?

2. Why were Crafter and Hunter domains chosen? Were there any specific properties of these domains that made them harder/easier for OCARL?

**Limitations:**

Yes

**Strengths And Weaknesses:**

Stengths:

1. The motivation is clear and well represented in the paper. The paper does a good job of placing their contributions in the context of existing works.

2. OCARL does generalize better than standard OORL approaches.

3. The methods section presents approach in detail, and should be reproducible for future researchers.

4. Ablation study is solid and gives a sense of which module helps with the performance bump.

Weaknesses:

1. Missing literature: The work cited below [A,B] seem closely related to some work on OOD generalization to novel combinations. In fact, experiments in [A] might also explain why OCMR helps generalize when using separate category-specific neural networks. Would be good to include a line or two discussing how OCMR connects to this work.

A. https://www.nature.com/articles/s42256-021-00437-5.pdf
B. https://www.nature.com/articles/s41593-018-0310-2

2. Might be good to clarify why the two domains mentioned in the paper were chosen.

---

> ### Author Response · Authors · 2022-08-01
> **Response**
>
> We thank the reviewer for the thoughtful comments.
> >Q1: Missing literature (A)(B)
>
> Thanks for your advice. We have added these papers to our citation list of the latest revision. We find (A) is more interesting, which says that the neural mechanism of specialization can lead to better generalization ability (We have added a sentence in Section3.3 to discuss (A) in our latest revision). This perspective also agrees with [1], which we have cited in our paper.
>
> [1] How modular should neural module networks be for systematic generalization
>
> >Q2: Might be good to clarify why the two domains mentioned in the paper were chosen.
>
> Crater and Hunter are chosen in our paper because of the consideration of both complexity and flexibility. Crafter is a complex domain with 19 kinds of objects in total and is of much more considerable significance to the RL community, which can be utilized to test OCARL's ability to handle complicated object combinations. On the other hand, Hunter is a flexible domain in which we have full control of the generating mechanism of different environment instances, making it possible for us to test OCARL's various properties (such as OOD generalization).
>
> >Q3: Is the number of classes (C) a known, required parameter? Was this known for the games, or not?
>
> In our current experiment setting, (C) is known and set by an oracle. However, it is easy to find a proper (C) automatically. For example, we can use a clustering method that does not need to specify (C) in advance (instead, the clustering method tells us the optimal (C)). Besides, many metrics (such as Silhouette Coefficient) exist to measure the quality of clustering and can be utilized to find a proper (C). For example, in Hunter, we can derive silhouette coefficients [0.648, 0.876, 0.961, 0.950, 0.942, 0.942, 0.923, 0.646, 0.633] for C=2,3,...,11, and automatically find C=4 is the most proper choice.
>
> >Q4: Why were Crafter and Hunter domains chosen? Were there any specific properties of these domains that made them harder/easier for OCARL?
>
> see Q2.

---

### Official Review · Reviewer_ZsUk · 2022-07-13

**Rating:** 6
**Confidence:** 4
**Soundness:** 3 good
**Presentation:** 3 good
**Contribution:** 3 good

**Summary:**

This paper proposed to learn object representations that can utiltize the category informations for perception and reasoning in reinforcement learning.
Experiments on two RL benchmarks show clear improvements of the proposed method over previous RL methods.
The ablation studies also cover a lot of aspects of the proposed method and show clear improvement of the each part of the proposed method.


**Questions:**


Q1: Some technical details about the SPACE method for unsupervised object discovery is not quite clear, I would

Q2: The unsupervised object discovery module is trained on images generated from a random policy, it would be interesting to see if using a better policy to generate the images used for training the object discovery module could improve the performance, for example, using the PPO policy to generate images.

Q3: Is there other tasks that are larger than Crafter and Hunter in terms of the image size and action space? I think given the impressive performance of the proposed method, it is reasonable to see if the proposed method can work on larger scale problems.

Q4: There are other object centric representation learning method that could be discussed, such as slot attention[R1].

[R1] Object-Centric Learning with Slot Attention, NeurIPS 2020


**Limitations:**

I think the major limitation of this paper is that the evaluated tasks are only two, I would expect to see more tasks given that the proposed method works so well the task tested.


**Strengths And Weaknesses:**


S1: The idea that object representations should encode category informations is interesting and IMO important for build generlizable RL methods.

S2: In the evaluation of this paper, the proposed method shows a very clear and strong improvements over previous methods.

W1: The tested benchmarks seems to be limited, I am wonderring if there are some larger scale benchmarks to test the proposed method?

W2: The description of the proposed method is not quite clear, I would recommand the paper to at least include some preliminary on SPACE in the main paper or the supplementary instead of asking the reader to refer to the original paper, IMO SPACE is an important component of the proposed method, so it is better to describe it in the main paper.

---

> ### Author Response · Authors · 2022-08-01
> **Response**
>
> Thanks for the thoughtful comments. We would like to clarify the concerns as follows:
> >W1: The tested benchmarks seem to be limited. I am wondering if there are some larger scale benchmarks to test the proposed method.
>
> There do exist many benchmarks in RL. However, most of them are not object-oriented or as complicated as Crafter. Perhaps NetHack can be a candidate, however its observations are symbolic (instead of pixel images) and NetHack needs training for more than 1 billion environment steps, which is beyond the interest of this paper. Crafter is actually very complicated and can 'benchmark the spectrum of agent capabilities' (as Figure4 shows), which recently attracted researchers' interest in the RL community. Crafter features a large object category number (19) and also a large possible action number (17), making it a challenging benchmark that is not well solved by other model-free RL algorithms.
>
> >W2: The description of the proposed method is not quite clear, I would recommend the paper to at least include some preliminary on SPACE in the main paper or the supplementary instead of asking the reader to refer to the original paper, IMO SPACE is an important component of the proposed method, so it is better to describe it in the main paper.
>
> Thanks for your advice. In our latest revision, we have added a section in the Appendix to describe SPACE in more detail. Actually, we have already introduced the inference model of SPACE (the most important part of SPACE which infers the object representations from raw images) in Section3.1, which should provide readers a rough understanding of SPACE.
>
> >Q1: Some technical details about the SPACE method for unsupervised object discovery is not quite clear
>
> See W2.
>
> >Q2: The unsupervised object discovery module is trained on images generated from a random policy, it would be interesting to see if using a better policy to generate the images used for training the object discovery module could improve the performance, for example, using the PPO policy to generate images.
>
> This is a good idea and also a promising direction for future work. However, in general, we do not have access to a well-trained policy to collect data from the environment, which means the UOD model may be incomplete (i.e., ignore some objects). This is exactly why we design OCAP, which makes OCARL robust to the UOD model's incompleteness.
>
> >Q3:  Is there other tasks that are larger than Crafter and Hunter in terms of the image size and action space? I think given the impressive performance of the proposed method, it is reasonable to see if the proposed method can work on larger scale problems.
>
> See W1.
>
> >Q4:  There are other object centric representation learning method that could be discussed, such as slot attention[R1]
>
> Thanks for your advice, and we added [R1] in our reference list and mentioned it in our related work. We think [R1] should fall into the spacial mixture models, because it iteratively clusters features that belongs to the same object.

---

> > ### Comment · Reviewer_ZsUk · 2022-08-04
> > **Thanks for the rebuttal**
> >
> > Thanks the author for the rebuttal, my concerns are resolved

---

### Author Response · Authors · 2022-08-01
**Revision**


We thank all the reviewers very much for their insightful comments and constructive suggestions to strengthen our work. After considering their suggestions, we have uploaded a revision of our paper. Here is the main change list:
- We added a sentence about how to choose a proper $C$ for unsupervised clustering (i.e., KMeans) in practice, which is one of the main concerns of Reviewer 1n95.
- We re-wrote the sentences in Section3.3 to make them clearer.
-  We added a paragraph to acknowledge the limitation suggested by Reviewer 2Frs.
-  We added a section in the Appendix to discuss SPACE in more detail as suggested by Reviewer ZsUk.
- Other minor changes.

---

### Meta-Review · Area_Chair_xPtv · 2022-09-08

**Recommendation:** Accept
**Confidence:** Certain

**Metareview:**

This paper received three positive reviews and one borderline reject. In the rebuttal, the negative reviewer did not propose a response, but the authors have given detailed responses to the problems. And the other reviewers did not propose further concerns. Thus, taking the comments of the reviewers into account, the AC decides to accept this paper.

**Award:**

No

---

### Decision · Program_Chairs · 2022-09-14

Accept